# Adherence to the Mediterranean Diet and Health-Related Quality of Life during the COVID-19 Lockdown: A Cross-Sectional Study including Preschoolers, Children, and Adolescents from Brazil and Spain

**DOI:** 10.3390/nu15030677

**Published:** 2023-01-29

**Authors:** Desirée Victoria-Montesinos, Almudena Tárraga-Marcos, Javier Brazo-Sayavera, Estela Jiménez-López, Héctor Gutiérrez-Espinoza, Josefa María Panisello Royo, Pedro J. Tárraga-López, José Francisco López-Gil

**Affiliations:** 1Faculty of Pharmacy and Nutrition, UCAM Universidad Católica San Antonio de Murcia, 30107 Murcia, Spain; 2Departamento de Ciencias Médicas, Facultad de Medicina, Universidad Castilla-La Mancha (UCLM), 02006 Albacete, Spain; 3PDU EFISAL, Centro Universitario Regional Noreste, Universidad de la República (UdelaR), Rivera 40000, Uruguay; 4Department of Sports and Computer Science, Universidad Pablo de Olavide (UPO), 41013 Seville, Spain; 5Health and Social Research Center, Universidad de Castilla-La Mancha, 16071 Cuenca, Spain; 6Escuela de Fisioterapia, Universidad de las Américas, Quito 170504, Ecuador; 7Fundación para el Fomento de la Salud, 28006 Madrid, Spain; 8Department of Environmental Health, Harvard University T.H. Chan School of Public Health, Boston, MA 02138, USA

**Keywords:** Mediterranean dietary patterns, eating healthy, lifestyle, pandemic, youths, children, adolescents, preschoolers

## Abstract

Scientific literature has suggested positive associations between the Mediterranean diet (MD) and the health-related quality of life (HRQoL) in young populations. However, to our knowledge, this relationship is unexplored during a situation of social isolation (i.e., lockdown). The objective of the current study is to examine the relationship between the MD and HRQoL during the COVID-19 lockdown among preschoolers, children, and adolescents from Brazil and Spain. This cross-sectional study includes a sample of 1099 three- to seventeen-year-old participants (47.6% girls) who were recruited via social networks. The HRQoL was assessed with the EQ-5D-Y. The Quality Index for Children and Teenagers (KIDMED) questionnaire was applied to evaluate the relationship between the MD and HRQoL. The highest prevalence of reported problems was found for worried, sad, or unhappy participants (39.8%). Furthermore, the lowest proportion of HRQoL problems was observed for “mobility” (2.5%). The proportion of high adherence to the MD was 44.3%. Participants with greater MD adherence reported higher HRQoL mean scores when compared with those who did not adhere to the MD (83.7 ± 0.6 vs. 85.6 ± 0.7, respectively; *p* < 0.05). Adherence to the MD and especially daily fruit intake were related to higher HRQoL during the COVID-19 lockdown among Brazilian and Spanish young people aged three to seventeen years.

## 1. Introduction

In March 2020, the World Health Organization (WHO) declared the global COVID-19 outbreak a pandemic [1]. An extreme increase in the number of infections led to the declaration of several restraint measures in most countries, such as lockdowns [2]. During these lockdowns, most countries (e.g., Brazil and Spain) established compulsory limitations, such as use of face masks, restricted international travel, and completely closed schools [1]. This social isolation, in combination with physical distancing, produced changes in the daily routines of citizens (e.g., physical activity levels, eating habits) [3,4]. For instance, a greater consumption of unhealthy foods was identified, especially among vulnerable populations such as children and adolescents [5]. Furthermore, a recent systematic review conducted by Pourghazi et al. [6] has pointed out uneven changes before and during the COVID-19 pandemic in young people (i.e., an increase in sweet and snack intake and a decrease in fruit, vegetables, and fast-food intake consumption). Supporting this notion, an additional aspect of concern from a public health perspective is the increase in the prevalence of malnutrition (e.g., excess weight) among children and adolescents during the COVID-19 pandemic [7]. Conversely, the COVID-19 lockdown has also lead to more family meals and a higher degree of resilience among family members [8], which has been related to greater psychological and physical health among this population [9]. 

Childhood and adolescence are periods characterized by an increase in food requirements due to higher nutritional demands for optimal physiological and psychological maturation [10]. Nutritional deficiencies among children and adolescents can lead to poorer development and future health problems [10]. In this line, the WHO states that healthy eating is a relevant factor in protecting against malnutrition and non-communicable diseases [11]. Among the healthiest dietary patterns, the Mediterranean diet (MD) is characterized by a dietary pattern rich in plant-based foods (fruits, vegetables, legumes, cereals, seeds, nuts, and olives), a moderate-to-high consumption of fish/seafood, a moderate intake of eggs, dairy products (preferably yogurt and cheese) and poultry, and a low intake of red meat, with olive oil as a main source of added fat [12]. Thus, this dietary pattern is one of the most well-known for its health benefits worldwide [13]. Although there is a belief that the MD is a characteristic eating habit of people living in countries around the Mediterranean Sea, other Mediterranean-type ecosystems can be placed from the 30° parallel to the 45° parallel of the southern or northern latitudes, with their coasts oriented towards the west [14]. Similarly, researchers from several non-Mediterranean countries (i.e., Colombia [15] and Brazil [16]) have adapted and validated the Mediterranean Diet Quality Index for Children and Teenagers (KIDMED) to assess adherence to the MD. The main reason for this is that this eating pattern is linked with a decrease in inflammatory status, as well as a reduced likelihood of being overweight and obese among young people [17,18], whereas a low adherence to the MD has been related to overweight and obesity in these age phases [19]. Despite its potential benefits, a low adherence to the MD seems to be prevalent in this population [20,21,22]. For instance, prior to the COVID-19 pandemic, the prevalence of children and adolescents having a low adherence to the Mediterranean diet was 4.2% in Spain [14], 14.9% and 27% for children and adolescents in Greece [15,16], and between 23.0% and 33.0% in Italy [17,18]. Based on these results, geographical region seems to play an important role in higher adherence, whereas Mediterranean countries do not always show the highest adherence to the Mediterranean diet. 

Another relevant factor from early childhood to adolescence is the health-related quality of life (HRQoL), which plays a key role in physical and psychological well-being [23]. HRQoL is defined as the individual’s perception of his or her position in life and the evaluation of different dimensions of this perception, along with their influence on health status [24]. Furthermore, HRQoL is a concept that has been widely documented in recent years to assess children and their health status in a comprehensive manner, including in its assessment physical, cognitive, social, and psychological functioning. It is crucial to prevent unhealthy behaviors [23]. Among young people, physical and psychological wellbeing declined significantly during the COVID-19 lockdown [25]. In fact, a recent systematic review by Nobari et al. [26] that included 3177 children and adolescents indicated a decrease in HRQoL among young people during the COVID-19 pandemic. Three articles showed that the COVID-19 pandemic significantly impacted the HRQoL of children and adolescents. Although another did not report a comparison between the pre-pandemic period and during the COVID-19 pandemic, a reduction in the HRQoL can be observed. Relevant HRQoL factors include dietary habits (i.e., frequency, composition, and amount of beverages/food consumed), which are associated with higher HRQoL scores in mental and physical dimensions [27]. In addition, these dietary habits have been associated with overweight and obesity in this population [28], which are related to lower HRQoL [29]. 

According to the results of some studies, the lockdown and physical distancing that occurred as a consequence of the COVID-19 pandemic may have contributed to a modification in some dietary habits and the HRQoL among young populations [30,31]. In this line, a systematic review by Della Valle et al. [4] indicated that adherence to the MD might have increased during COVID-19 lockdown. Furthermore, some systematic reviews have concluded that the COVID-19 pandemic has led to a decline in the HRQoL of children and adolescents [26,32]. A recent systematic review analyzed the association of adherence to the MD with the HRQoL in young people [27]. However, all studies included in that review were conducted before [33,34,35,36] and other was performed after [37] the COVID-19 lockdown, indicating an association between a greater adherence to the MD and a higher HRQoL among young populations [27,37]. To our knowledge, this is the first study that has analyzed this relationship in young people during the situation of social isolation (i.e., lockdown) due to the COVID-19 pandemic. Thus, the objective of the current study is to examine the association between the MD (i.e., adherence to this eating pattern and its specific components) and HRQoL during the COVID-19 lockdown among preschoolers, children, and adolescents aged 3−17 years from Brazil and Spain.

## 2. Materials and Methods

### 2.1. Participants and Study Design

Out of the 1263 participants from Brazil and Spain, 143 were eliminated since they were too young (aged < 3 years) or too old (aged > 17 years) for the study. Additionally, 21 participants were removed due to missing data. This left 1099 participants whose data were included in the final analysis. As in-person contact was not allowed, these participants were recruited through social networks. Through a snowball sampling technique, an online survey was designed and delivered in both Brazil and Spain. The survey took approximately 15 min to complete and was filled out by the participants’ parents or guardians. Before carrying out the survey, the objective of this study was explained and informed consent was obtained. The data was collected over a period of 15 days in both countries (from 29 March to 13 April 2020). During this period, the entire Brazilian and Spanish populations were expected to remain at home (excepting essential workers) and were only allowed to go out for healthcare, basic food shopping, and some justified exceptions. As criteria, only parents/legal guardians of the young Brazilian and Spanish populations aged 3–17 years who signed the informed consent were involved. Regarding the exclusion criteria, participants were not included when their parents/guardians did not totally complete the online survey. This study had the approval of the ethical committees of the *Universidade Tecnologica do Paraná* (CAAE: 32023220.8.0000.5547; approval number: 4.275.232 and the *Universidad Católica de Murcia* (code: CE112001).

### 2.2. Procedures

#### 2.2.1. Health-Related Quality of Life (Dependent Variable)

The HRQoL (health-related quality of life) was evaluated by the EQ-5D-Y (proxy version 1), a tool specifically designed for evaluating HRQoL in young people [38]. The EQ-5D-Y assesses five different aspects of HRQoL (mobility, self-care, ability to carry out usual activities, presence of pain or discomfort, and emotional well-being) on a three-point scale (no problems, some problems, or a lot of problems). The results of the EQ-5D-Y are represented by a three-digit code, with each digit representing the level of severity in each of the five dimensions. The dimensions of HRQoL were also grouped into two categories: “no problems” or “any problems” (combining “some problems” and “a lot of problems”). In addition, the EQ-5D-Y includes a visual analogue scale (VAS) with scores ranging from 0 (the lowest HRQoL) to 100 (the highest HRQoL). The reliability and validity of the EQ-5D-Y have been established in the previous research [39]. 

#### 2.2.2. Adherence to the Mediterranean Diet (Independent Variable)

Adherence to the Mediterranean diet (MD) was evaluated using the KIDMED [40]. The KIDMED is a sixteen-question test that ranges from −4 to 12 points. All the items included in the KIDMED and its scoring system can be found in Appendix A. Questions about unhealthy aspects of the MD were scored as −1 (e.g., “Takes sweets and candy several times every day”), and questions about healthy aspects were scored as +1 (e.g., “Takes a fruit or fruit juice every day”). The total scores from the KIDMED were then divided into three categories: high MD (≥ 8 points), moderate MD (4–7 points), and low MD (≤ 3 points).

#### 2.2.3. Covariates

The following information was collected from the parents or guardians of the participants: age, sex (male or female), nationality, educational level of the primary breadwinner (non-university or university), socioeconomic status, excess weight (overweight or obese), and daily movement behaviors. These variables were all considered to be covariates. The family affluence scale-III (FAS-III) score ranges from 0 to 13 points, with higher scores indicating a higher socioeconomic status based on the responses to six questions about the family’s resources (e.g., number of vehicles, bedrooms, computers, etc.) [41]. Anthropometric data (e.g., height, weight) were also collected for the children, and the body mass index (BMI) z-score and the proportion of excess weight (i.e., overweight and obesity) were computed following the World Health Organization (WHO) criteria [42,43]. Daily movement behaviors, including physical activity, recreational screen time, and sleep duration, were also measured. Physical activity was evaluated by asking parents how many days their child was physically active for at least 60 min in the previous week. Recreational screen time was measured by asking parents about the time their child spent on various sedentary activities (e.g., watching TV, playing video games, or using electronic devices) as follows: (a) “How many hours a day, during the COVID-19 lockdown, does your child usually spend using electronic devices such as computers, tablets or smartphones for other purposes (e.g., homework, emailing, tweeting, Facebook, chatting, surfing the internet)?”; (b) (b) “How many hours a day, during the COVID-19 lockdown, does your child spend playing games on a computer, games console, tablet, smartphone or other electronic device (not including moving or fitness games)?”; (c) “How many hours a day, during the COVID-19 lockdown, does your child spend watching TV, videos (including YouTube or similar services), DVDs, and other entertainment on a screen?”. The three answers were summed considering a week distribution of five weekdays and two weekend days. Sleep duration was determined by asking parents about the bedtimes and wake times (for weekdays and weekend days separately) of their child as follows: “What time does your child usually go to bed?” and “What time does your usually get up?”. The average daily sleep duration was computed for each participant as follows: [(average nocturnal sleep duration on weekdays × 5) + (average nocturnal sleep duration on weekends × 2)]/7.

### 2.3. Statistical Analysis

The descriptive data for the study were presented in two ways: as the mean and standard deviation in the case of continuous data, and as the number and percentage in case of categorical data. Differences between adherence to the MD and problems in the HRQoL domains were tested using the Pearson’s chi-square test (*χ*^2^) or Fisher’s exact test (when expected values in any of the cells of a contingency table were below five participants). Preliminary analyses showed no interaction between adherence to the MD and sex (*p* = 0.351) or age group (*p* = 0.174) in relation to HRQoL. Thus, all the samples were analyzed together to increase the statistical power. Analyses of covariance (ANCOVAs) were performed to assess the differences in HRQoL in relation to adherence to the MD, as well as to determine the specific MD components associated with HRQoL. Furthermore, linear regression analyses were performed to test the individual MD components associated with a higher HRQoL. Sex, age, nationality, socioeconomic status, breadwinner’s educational level, excess weight, and daily movement behaviors were included as covariates. Statistical analyses were performed by the software Statistical Package for Social Sciences (SPSS, IBM Corp., Armonk, NY, USA) in its version 28 for Windows. A *p*-value < 0.05 denoted statistical significance.

## 3. Results

Table 1 shows the descriptive data for study participants. The proportion of high adherence to the MD was 44.3%. The VAS mean score was 84.5 ± 15.9. The lowest proportion of HRQoL problems was observed for “mobility” (2.5%). Conversely, the highest proportion was found for “feeling worried, sad, or unhappy” (39.8%).

In Table 2, we assessed the association between adherence to the MD and the different domains of HRQoL. Despite a higher proportion of problems across all HRQoL domains in participants with low/moderate MD, only significant differences were obtained for the “having pain or discomfort” domain (*p* = 0.020). Sub-group analyses by sex and age group can be found in Appendix A.

Figure 1 depicts the VAS mean score in relation to adherence to the MD. After adjusting by several covariates, participants with high MD adherence reported greater HRQoL mean scores (M = 83.7; SE = 0.6) than those participants with low/moderate MD adherence (M = 85.6; SE = 0.7). In both the unadjusted and adjusted models, the differences between groups were statistically significant (*p* < 0.05 for both). Appendix A show sub-group analyses by sex and age group, respectively.

Table 3 indicates the association between different MD components of the KIDMED and the HRQoL scores. After adjusting by several covariates, eating fruit or consuming fruit juice every day was significantly associated with a greater HRQoL score (*p* < 0.005).

## 4. Discussion

To our knowledge, this is the first study to analyze associations between the MD (overall adherence and its specific components) and the HRQoL in preschoolers, children, and adolescents from Brazil and Spain during the COVID-19 lockdown. Overall, our results showed that a higher adherence to the MD was related to higher levels of the HRQoL and lower instances of HRQoL problems in the “having pain or discomfort” domain. When the different MD components were considered, daily fruit intake or fruit use was associated with higher scores of HRQoL.

Our findings depict that a high adherence to the MD was linked with higher HRQoL scores. This result is in accordance with most of the studies conducted on this matter, both before [27] and after the COVID-19 lockdown [44]. Nevertheless, divergent results have also been found [27]. These divergent results could be explained by the age of the included samples, with different levels of psychological maturity [45], or by the consumption of different proportions of food groups [46]. Although the mechanism by which the MD may improve the HRQoL is not fully explained, it could be related to the high amount of antioxidants, polyphenols, vitamins, and dietary fiber present in some foods (i.e., fruits). The daily intake of fruit (an eating habit characteristic of the MD) could lead to benefits in mental well-being and normal brain function [47,48]. An additional possible reason for the association between the MD and HRQoL could be (at least partially) explained due to the possible beneficial influence of a healthy diet on certain psychological factors (i.e., positive affect, life satisfaction, moods, and emotions) [49,50]. Another possible explanation could be related to the association between eating habits, sleep-related problems, and the HRQoL. In this sense, López-Gil et al. [51] observed a relationship among healthy eating habits and sleep-related problems, which has been related to a lower HRQoL [52]. Their results showed that eating habits characteristic of the MD (i.e., a higher fruit, vegetable, and bean intake, a lower sweet consumption, having breakfast more often, and having family meals) were related to lower sleep-related problems. It is possible that the consumption of certain MD components facilitates an adequate sleep duration (diminishing sleep-related problems) and, therefore, a higher HRQoL [53,54]. Lastly, breakfast (an important component of the MD) has been associated with greater psychosocial health in Spanish preschoolers, children, and adolescents [55,56]. In this sense, social context appears to be crucial in this relationship [57] as family meals at home may help to increase the quality of the meals, which could improve well-being and decrease psychosocial behavioral problems, increasing the HRQoL in this population [58].

In addition, another finding of the current study was that a higher adherence to the MD was related to lower pain or discomfort symptoms during the COVID-19 lockdown (especially in children and adolescents). Previous studies have depicted that a higher adherence to the MD is related to lesser pain in children and adolescents [59,60]. Although the mechanisms by which a healthy diet may increase the HRQoL are unknown, the intake of some type of foods (e.g., fruits, vegetables, and olive oil) typical of the MD could offer benefits against oxidative stress and inflammation [61,62], which have been hypothesized to be pain contributors [63,64]. Therefore, it is possible that participants with a higher adherence to the MD have lower levels of oxidative stress and inflammation that could contribute to perceiving less pain.

On the other hand, by specifically analyzing the components of the MD, our results showed that a daily consumption of fruit/fruit juice was related to a higher HRQoL. Despite being a period characterized by a decrease in healthy behaviors [3,65,66,67], young people who had this healthy eating habit reported a higher HRQoL in comparison with those who did not. These results are in line with those obtained in different cross-sectional pre-pandemic studies [68,69,70], in which a higher fruit consumption was related to a higher HRQoL. This finding could be explained due to the antioxidant and anti-inflammatory effects of the bioactive compounds contained in fruits such as folate, vitamin C, polyphenols, fiber, etc. [71]. In addition to physical health, the adequate consumption of fruits and their biocomponents also seems to improve mental health [48]. This may be because of the role of essential nutrients provided by healthy food (i.e., fruits) in neurotransmitter formation related to mental health, such as serotonin, dopamine, and oxytocin [72], which could lead to a higher HRQoL.

This study has some strengths that should be acknowledged, including the use of large samples of preschoolers, children, and adolescents from Brazil and Spain, and being the first to examine the specific association of the MD and its specific components with HRQoL during the COVID-19 lockdown in this age group. Conversely, this study has several limitations that should be taken into account. First, the cross-sectional design of the study means that it is not possible to determine cause-and-effect associations. Thus, we do not have information from prior to the COVID-19 lockdown on some variables that could influence the results obtained (e.g., previous adherence to the MD or previous diets). For a better understanding of the link between adherence to the MD and the HRQoL, it would be necessary to conduct longitudinal studies. Second, the information obtained was based on self-reported questionnaires that may introduce social desirability and recall biases. Third, the KIDMED does not provide exact information on types of food and the frequency of consumption. This means that, for example, in evaluating the consumption of dairy products, we do not know whether skimmed or whole milk was consumed, which may influence neurotransmitter formation such as serotonin and dopamine, since whole milk has a higher quantity of these precursors (tryptophan and tyrosine) [73]. Fourth, the compulsory limitations during the COVID-19 pandemic could differently influence the adherence to the MD and the HRQoL according to the type of lockdown imposed by each country’s government (e.g., a total or partial lockdown). Fifth, although several covariates were adjusted, residual confounding is still possible (e.g., type of family and frequency of family meals).

## 5. Conclusions

Adherence to the MD, especially daily fruit intake, were related to higher HRQoL scores during the COVID-19 lockdown among Brazilian and Spanish young people aged 3 to 17 years. This finding is clinically significant, as the pandemic appears to have deteriorated the HRQoL in the young population and a better understanding of the associated factors (i.e., healthy diet) could help to establish concrete measures in case of situations of social isolation. Further studies with different designs (i.e., interventions) are required to establish cause-and-effect relationships and to verify whether greater adherence to the MD produces improvements in the HRQoL in the young population.

## Figures and Tables

**Figure 1 nutrients-15-00677-f001:**
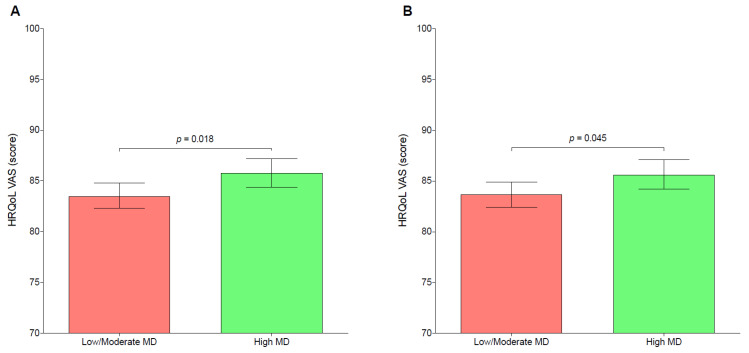
Mean differences of health-related quality of life according to the degree of adherence to the Mediterranean diet during the COVID-19 lockdown. HRQoL—health-related quality of life; MD—Mediterranean diet; VAS—visual analogue scale. (**A**): unadjusted; (**B**): adjusted by sex, age, nationality, socioeconomic status, breadwinner’s educational level, excess weight, and daily movement behaviors. *p* values obtained by analysis of covariance.

**Table 1 nutrients-15-00677-t001:** Characteristics of the study participants (N = 1099).

Variables	M/n	SD/%
Age (years)	11.5	4.5
Preschoolers (3–5 years)	146	13.3
Children (6–12 years)	444	40.4
Adolescents (13–17 years)	509	46.3
Sex		
Boys	576	52.4
Girls	523	47.6
Nationality		
Brazilian	495	45.0
Spanish	604	55.0
Breadwinner’s educational level		
University studies	538	49.0
Non-university studies	561	51.0
Socioeconomic status		
FAS-III (score)	7.5	2.3
Anthropometric data		
Weight (kg)	43.0	19.5
Height (cm)	145.5	24.8
BMI (z-score)	0.9	2.0
Overweight/Obesity ^a^	467	42.5
Daily movement behaviors		
Physical activity (days/week)	4.0	2.3
Recreational screen time (h/day)	7.6	5.9
Sleep duration (h/day)	9.9	1.5
Adherence to the MD		
KIDMED (score)	7.0	2.4
Low/Moderate MD	612	55.7
High MD	487	44.3
HRQoL		
VAS (score)	84.5	15.9
Mobility (%, any problem)	28	2.5
Looking after myself (%, any problem)	112	10.2
Doing usual activities (%, any problem)	113	10.3
Having pain or discomfort (%, any problem)	181	16.5
Feeling worried, sad, or unhappy (%, any problem)	437	39.8

Data expressed as a mean and standard deviation for continuous variables or numbers and percentage for categorical variables. BMI—body mass index; FAS—family affluence scale; HRQoL—health-related quality of life; KIDMED—Mediterranean Diet Quality Index for Children and Teenagers; MD—Mediterranean diet; and VAS—visual analogue scale. ^a^ According to the World Health Organization criteria [43].

**Table 2 nutrients-15-00677-t002:** Association between Mediterranean diet adherence and problems in different dimensions of health-related quality of life.

HRQoL Domain ^a^	Low/Moderate MD	High MD	*p*
Mobility	18 (2.9)	10 (2.1)	0.442
Looking after myself	62 (10.1)	50 (10.3)	0.941
Doing usual activities	71 (11.6)	42 (8.6)	0.106
Having pain or discomfort	115 (18.8)	66 (13.6)	0.020
Feeling worried, sad, or unhappy	246 (40.2)	191 (39.2)	0.743

^a^ Data expressed as proportion of participants reporting any problem in the different health-related quality of life domains, according to adherence to the Mediterranean diet. HRQoL—health-related quality of life; MD—Mediterranean diet. *p* values obtained by Pearson’s chi-square test or Fisher’s exact test.

**Table 3 nutrients-15-00677-t003:** Association between different Mediterranean diet components and health-related quality of life score.

Items	*B*	SE	LLCI	ULCI	*p*
Takes a fruit or fruit juice every day	3.59	1.28	1.09	6.09	0.005
Has a second fruit every day	0.29	1.13	−1.94	2.51	0.800
Has fresh or cooked vegetables regularly once a day	−0.05	1.20	−2.41	2.31	0.968
Has fresh or cooked vegetables more than once a day	−0.04	1.15	−2.29	2.21	0.973
Consumes fish regularly (at least 2–3 times per week)	−1.84	1.14	−4.07	0.39	0.106
Goes more than once a week to a fast-food (hamburger) restaurant	0.99	1.55	−2.05	4.02	0.522
Likes pulses and eats them more than once a week	0.00	1.41	−2.78	2.77	0.999
Consumes pasta or rice almost every day (5 or more times per week)	−0.35	1.19	−2.68	1.98	0.765
Has cereals or grains (bread, etc.) for breakfast	−0.06	1.20	−2.41	2.28	0.958
Consumes nuts regularly (at least 2–3 times per week)	−0.13	1.06	−2.22	1.96	0.905
Uses olive oil at home	−0.76	1.75	−4.20	2.68	0.665
Skips breakfast	−2.28	2.15	1.94	−6.49	0.289
Has a dairy product for breakfast (yogurt, milk, etc.)	−0.36	1.54	−3.37	2.66	0.816
Has commercially baked goods or pastries for breakfast	−0.59	1.45	−3.43	2.26	0.686
Takes two yogurts and/or some cheese (40 g) daily	0.10	1.00	−1.86	2.06	0.921
Takes sweets and candy several times every day	0.48	1.23	−1.94	2.90	0.698

Data expressed as non-standardized beta coefficient, standard error, and 95% confidence interval. Adjusted by sex, age, nationality, socioeconomic status, breadwinner’s educational level, excess weight, and daily movement behaviors. *p* values obtained by multiple linear regression.

## Data Availability

Not applicable.

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
