# Peer review of "Adherence to the Mediterranean Diet and Health-Related Quality of Life during the COVID-19 Lockdown: A Cross-Sectional Study including Preschoolers, Children, and Adolescents from Brazil and Spain"

_nutrients, 2023, doi:10.3390/nu15030677_

Round 1

Reviewer 1 Report

Dear Athors,

Congratulations to the authors for undertaking a study related to the use of a widely recognized diet as health-promoting (MD) and health-related quality of life during the COVID-19 lockdown.

Below are my suggestions / comments:

In the title, "young people" does not reflect the study population. Children as young as 3 years old are not associated with this term. "Children and adolescents" would be better.

There is no information on what statistical test was used to compare the values in Table 2 and Figure 1.

Describe what was the adjustment of the HRQoL VAS scores by the variables listed (Figure 1)?

Please describe the selection of variables for the model Table 3. It appears that only one variable "Takes a fruit or fruit juice every day" is statistically significant in the model. Variables that are not statistically significant have no effect on the model.

There is no separate "Conclusions" section with clear conclusions drawn from the study analysis.

Author Response

Thank you for your feedback.
Point 2. Below are my suggestions / comments: In the title, "young people" does not reflect the study population. Children as young as 3 years old are not associated with this term. "Children and adolescents" would be better.
- Following your comment, we have replaced with “preschoolers, children, and adolescents”. Thank you.
Point 3. There is no information on what statistical test was used to compare the values in Table 2 and Figure 1.
- We have included this information in the table/figure captions. Thank you.
Point 4. Describe what was the adjustment of the HRQoL VAS scores by the variables listed (Figure 1)?
- We have described the adjustment variables in the Statistical analysis section as follows: “Sex, age, nationality, socioeconomic status, breadwinner’s educational level, excess weight, and daily movement behaviors were included as covariates.” Additionally, this information appears in the figure legend. Thank you.
Point 5. Please describe the selection of variables for the model Table 3. It appears that only one variable "Takes a fruit or fruit juice every day" is statistically significant in the model. Variables that are not statistically significant have no effect on the model.
- Thank you for your comment. We included all the items of the KIDMED index. Therefore, only the item “Takes a fruit or fruit juice every day” was statistically significant. Thank you.
Point 6. There is no separate "Conclusions" section with clear conclusions drawn from the study analysis.
- We have included a “conclusions” section. Thank you.

Reviewer 2 Report

Thank you for the article, MD is considered for many years now as being the healthiest diet and the easiest to follow. I wonder how many people in Mediteraneean areas are really still following it, since "modern" types of diets , not at all healthy, have taken its place. Even more, I wonder if in Northern and Central Spain, with heavy consumption of red meat and cured meat products, can qualify as MD areas. Also Brazil?!I did not knew that they eat and MD diet, please give some references and explain.

The study has some drawbacks:

- you investigated children form 3, to 17. The gap is huge in every area of life and development in this interval You really cannot compare a 4 year old with a 16, which, in our days, is almost an adult. Please, try and split the analyzis on group ages

- you do not know the adherence to MD BEFORE covid. Maybe they never followed it and nothing has changed during lockdown. Maybe try to find more references showing some percents for the ages you took into consideration. 

- we do not know what KIDMED is and the reader maybe cannot and has not the tome to access references, so please present it in a more detailed manner

- maybe a description of what happened in the 2 countries during covid is useful. We do not know. I think in Spain there were very strict lockdowns for a longer period of time. Do not know anything about Brazil. These are very important things, taking their toll on the accuracy of the results. 

- in conclusions you say that the present study will help us in future lockdowns. Please, think again and reformulate...do we all expect such a thing in future?

Author Response

Thank you for your time and review. Although consumer habits are changing, Spain is still a Mediterranean country (due to its geographical location). It is also able to provide the typical Mediterranean diet foods that have been associated with numerous health benefits (Guasch‐Ferré & Willett, 2021). Similarly, there are other non-Mediterranean countries that also analyze adherence to this dietary pattern, as one of the healthiest eating patterns that exists (Bach-Faig et al., 2011). Because in other geographic locations, adherence to this dietary pattern is still possible (Köppen, 1936). In Brazil, the most widely used questionnaire to assess adherence to the Mediterranean Diet (KIDMED) has been validated for use in children and adolescents (Simon et al., 2020). We have tried to improve the context of the use of this tool in a non-Mediterranean country in the manuscript.

Point 2. The study has some drawbacks: you investigated children form 3, to 17. The gap is huge in every area of life and development in this interval You really cannot compare a 4 year old with a 16, which, in our days, is almost an adult. Please, try and split the analyzis on group ages

- We agree with the reviewer. Responses could be different according to age group or sex. However, we found no interaction between these variables and adherence to the Mediterranean diet in relation to health-related quality of life. We have included this information as follows: “Preliminary analyses showed no interaction between adherence to the Mediterranean diet and sex (p=0.351) or age group (p=0.174) in relation to HRQoL. Thus, all the sample was analyzed together to increase the statistical power”. Nevertheless, we have included subgroup analyses as a supplementary material following your suggestion (Table S2, Fig. S1, Fig. S2). Thank you.

Point 3. You do not know the adherence to MD BEFORE covid. Maybe they never followed it and nothing has changed during lockdown. Maybe try to find more references showing some percents for the ages you took into consideration.

- It is true that we do not know the adherence to the Mediterranean diet before COVID-19 lockdown. However, this is a cross-sectional study, and it is a limitation inherent to this type of studies. Similarly, our aim was not to evaluated differences before and during COVID-19 lockdown. We tried to examine the relationship between Mediterranean diet and health-related quality of life during the COVID-19 lockdown among preschoolers, children, and adolescents from Brazil and Spain. Although the COVID-19 lockdown could affect the health-related quality of life of the participants, our results show that those who had a high adherence to Mediterranean diet had a higher quality of life. Thank you.

Point 4. We do not know what KIDMED is and the reader maybe cannot and has not the tome to access references, so please present it in a more detailed manner.

- Thank you for your comment. In addition to the description of the KIDMED in the Methods section, we have included a supplementary table (Table S1) indicating the different items included in the KIDMED index and its scoring system.

Point 5. Maybe a description of what happened in the 2 countries during covid is useful. We do not know. I think in Spain there were very strict lockdowns for a longer period of time. Do not know anything about Brazil. These are very important things, taking their toll on the accuracy of the results.

- Even though the two governments' responses to the COVID-19 pandemic were inconsistent, in the period analyzed the situation was similar in both countries. We have tried to explain this in the introduction section: “During these lockdowns, most of countries (e.g., Brazil, Spain) established compulsory limitations, such as use of face masks, restricted international travel, and completely closed schools (World Health Organization, 2020)”. Similarly, we have also added this information in the methods section: “The data was collected over a period of 15 days in both countries (from March 29 to April 13, 2020). During this period, the entire Brazilian and Spanish population should remain at home (excepting essential workers) and was only allowed to go out for healthcare, basic food shopping, and some justified exceptions”. As shown, preschoolers, children, and adolescents from both countries were at home, in a very similar lifestyle situation. In addition, we have included a further limitation as follows: “Fourth, the compulsory limitations during the COVID-19 pandemic could influence differently adherence to the MD and HRQoL according to the type of lockdown imposed by each country’s government (e.g., total or partial lockdown)”. Thank you.

Point 6. In conclusions you say that the present study will help us in future lockdowns. Please, think again and reformulate...do we all expect such a thing in future?
- Hopefully there will be no more lockdowns. We have replaced with “in case of situations of social isolation”.

References
Bach-Faig, A., Berry, E. M., Lairon, D., Reguant, J., Trichopoulou, A., Dernini, S., Medina, F. X., Battino, M., Belahsen, R., Miranda, G., & Serra-Majem, L. (2011). Mediterranean diet pyramid today. Science and cultural updates. Public Health Nutrition, 14(12A), 2274–2284. https://doi.org/10.1017/S1368980011002515
Guasch‐Ferré, M., & Willett, W. C. (2021). The Mediterranean diet and health: A comprehensive overview. Journal of Internal Medicine, 290(3), 549–566. https://doi.org/10.1111/joim.13333
Köppen, V. (1936). Das geographische System der Klimate. Gebrüder Borntraeger.
Simon, M. I. S. dos S., Forte, G. C., & Marostica, P. J. C. (2020). Translation and cultural adaptation of the mediterranean diet quality index in children and adolescents. Revista Paulista de Pediatria, 38. https://doi.org/10.1590/1984-0462/2020/38/2018242
World Health Organization. (2020). Coronavirus disease 2019 (COVID-19). Situation report – 51. https://www.who.int/docs/default-source/coronaviruse/situation-reports/20200311-sitrep-51-covid-19.pdf?sfvrsn=1ba62e57_10

Reviewer 3 Report

1) Title and introduction are clear. The background to the topic and research gap has been adequately described.

2) Study design is appropriate. Variables, covariates and statistical analysis have been clearly articulated. 

3) Table 1 - add age range for each of the groups i.e preschoolers, children & adolescents. 

4) Table 2, Fig 1 - it's worth doing further sub-group analysis according to age groups; perhaps add this as supplementary docs to add more value to the analysis done. 

5) Discussion is fair, with acknowledgement to study limitations/assumptions. 

Author Response

Thank you so much for your feedback.
Point 2. Study design is appropriate. Variables, covariates and statistical analysis have been clearly articulated.
- We appreciate your evaluation. Thanks.
Point 3. Table 1 - add age range for each of the groups i.e preschoolers, children & adolescents.
- Done. Thank you.
Point 4. Table 2, Fig 1 - it's worth doing further sub-group analysis according to age groups; perhaps add this as supplementary docs to add more value to the analysis done.
- We have added sub-group analyses according to the age group. This information has been included as supplementary material (Table S1 and Figure S1). Thank you.
Point 5. Discussion is fair, with acknowledgement to study limitations/assumptions.
- Thank you for your comment.

Round 2

Reviewer 2 Report

I think you answered adequately to my request. All the questions were answered.

However, I think that there is a little bit of a stretch to state that your QoL is better in lockdown just because you eat Mediterranean. Just during that period, nobody knows what you ate before...because this is a cross sectional study!

Yes, statistics show the connection. You can feed whatever you want in statistics and results will come out.  I do not deny that Mediterranean diet is healthy, but this comes , like in any other diet, after a long time. Not just after a few days/month of covid... Inflammation, body weight change in time. In introduction (line 80 and fw) you quote yourself low percent of adherence to MD in youngsters.  So we can presume that the sample you worked on was not very adherent to MD before...

Sincerely, I think that the connection high adherence/ high QoL during covid is due to many other factors, maybe like family ambiance and more traditional M habits in the household, better sustaining the coping mechanisms of the children during hard times . 

Overall, the article is correct, but for me the confirmation of the hypothesis, without knowing older dietary habits of the children is rather unconvincing. You do not have a better QoL just by changing your diet.

Please, insert more about eventual factors like previous diets or other elements that could influence your results, not just that this is cross sectional and longitudinal are needed. It seems too superficial to dismiss al pletora of factors linked with a better QoL just in a sentence. 

Thank you
